# Towards Enabling Blind People to Fill Out Paper Forms with a Wearable Smartphone Assistant

Shirin Feiz*     Anatoliy Borodin†     Xiaojun Bi‡     IV Ramakrishnan §

Stony Brook University

## ABSTRACT

We present PaperPal, a wearable smartphone assistant which blind people can use to fill out paper forms independently. Unique features of PaperPal include: a novel 3D-printed attachment that transforms a conventional smartphone into a wearable device with adjustable camera angle; capability to work on both flat stationary tables and portable clipboards; real-time video tracking of pen and paper which is coupled to an interface that generates real-time audio read outs of the form's text content and instructions to guide the user to the form fields; and support for filling out these fields without signature guides. The paper primarily focuses on an essential aspect of PaperPal, namely an accessible design of the wearable elements of PaperPal and the design, implementation and evaluation of a novel user interface for the filling of paper forms by blind people. PaperPal distinguishes itself from a recent work on smartphone-based assistant for blind people for filling paper forms that requires the smartphone and the paper to be placed on a stationary desk, needs the signature guide for form filling, and has no audio read outs of the form's text content. PaperPal, whose design was informed by a separate wizard-of-oz study with blind participants, was evaluated with 8 blind users. Results indicate that they can fill out form fields at the correct locations with an accuracy reaching 96.7%.

**Index Terms:** Human-centered computing—Accessibility—Accessibility technologies—; Human-centered computing—Human computer interaction (HCI)

## 1 INTRODUCTION

Paper documents continue to persist in our daily lives, notwithstanding the paperless digitally connected world we live in. People still continue to encounter paper-based transactions that require reading, writing and signing paper documents. Examples include paper receipts, mails, checks, bank documents, hospital forms and legal agreements. A recent survey shows that over 33% of transactions in organizations are still done with paper documents [4]. Many of these paper documents, at the very least, require affixing signatures on them. While it is straightforward for sighted people to write and affix their signatures on paper, for people who are blind this is challenging, if not impossible to do independently. When it comes to writing, blind people invariably rely on sighted people for assistance. Such assistance may not always be readily available, but more troublingly, having to depend on others for writing always comes with a loss of privacy. To make matters worse, unlike reading assistants for blind people, of which there are quite a few (e.g., [2, 8]), there are hardly any computer-assisted aids that can help them to write on paper independently, a problem that has taken on added significance due to the recent pandemic-driven upsurge in mail-in balloting. In fact, a recent lawsuit was brought by blind plaintiffs on the discriminatory

---

*e-mail: sfeizdisfani@cs.stonybrook.edu
†e-mail: anatoliy.borodin@stonybrook.edu
‡e-mail: xiaojun@cs.stonybrook.edu
§e-mail: ram@cs.stonybrook.edu

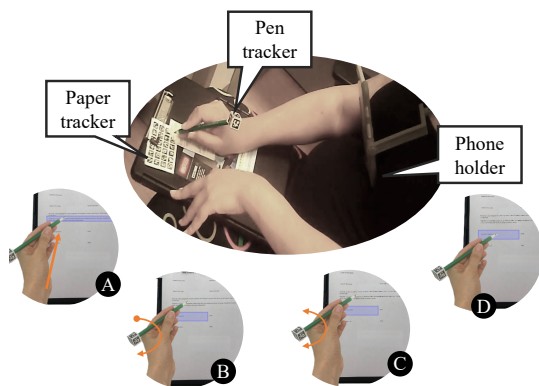

Figure 1: A blind user filling out a form using PaperPal. An interaction scenario: (A) The pen is pointing to a text item and is read out, (B) Rotate the pen right to read the next item, (C) Bi-directional rotate to navigation to the form field, (D) Fill out the field.

nature of mail-in paper ballots since they could not be filled out without compromising confidentiality [5].

There are two essential aspects to a form-filling assistant for blind people: (1) document annotation which includes capturing the image of the document with a camera, and automatic identification of all its items, namely, text segments, form fields and their labels; and (2) the design and implementation of an interface to enable blind people to access and read all the items of the document and fill out the fields independently.

In so far as (1) is concerned, the existence of several smartphone reading apps for them (e.g., SeeingAI [8], KNFB reader [2], and voice dream scanner [26]) has established the feasibility of acquiring images of paper documents by blind people using a smartphone. These apps demonstrate that blind users can independently use their audio interface to capture the image of the document. The aforementioned apps also extract text segments from the captured images using OCR and document segmentation methods, which are then read out to the user. In so far as forms are concerned, it is possible to extract form fields and their labels from document images using extant vision-based systems such as Adobe [10], and AWS Textract [9]. In contrast, the HCI aspects of an interface for a form-filling assistant for blind people is a challenging and relatively understudied research problem, and is the primary focus of this paper.

Of late, research on HCI aspects of writing aids for blind people is beginning to emerge. A recent work describes a first-of-its-kind smartphone-based writing-aid, called WiYG, for assisting blind people to fill out paper forms by themselves [28]. WiYG uses a 3D printed attachment to keep the phone upright on a flat table and redirects the focus of the phone's camera to the document that is placed in front of the phone. The paper and phone in WiYG are kept stationary; The user receives audio instructions to slide the signature guide – a card similar in size to regular credit cards that has a rectangular opening in the middle to help blind people sign on

papers – to different form fields on the paper. All the form fields are manually annotated apriori. In addition, visual markers are affixed to the signature guide for tracking its locations with the camera.

WiYG work has opened up new design questions and challenges that could form the basis for next generation computer-aided paper-form-filling writing assistants for blind people. We explore some of these questions here. Firstly, WiYG provides no readouts of the text in the form documents which arguably is desirable, especially for documents that require signatures. Secondly, WiYG simply steps through each form field in the document one by one without backtracking. In practice one would like to seamlessly switch back and forth between the fields and fill them in any order. Thirdly, WiYG requires a flat table to keep the paper as well as the phone stationary during use. The ability to operate in different situational contexts such as documents on non-stationary portable surfaces such as clipboards makes for a more flexible computer-aided reading/writing *wearable* assistant. In fact, often times blind users find themselves in situations where the documents they are asked to review and sign such as forms at hospitals and doctor's offices are on clipboards.

To explore these questions we employed a user-centered design approach. We started with a Wizard of Oz (WoZ) pilot study with eight blind participants to understand the feasibility of filling paper forms on a clipboard. The study included paper forms placed on both flat desks and portable clipboards with the wearable cameras worn over the chest or attached to glasses, to mimic smart glasses. The study was designed to elicit data on several key questions including: (1) How do blind people write on paper attached to a portable clipboard? (2) Where can the camera be worn conveniently and in a way that the pen and paper are visible within the camera's field of view? (3) Considering all the camera-clipboard movements, how can blind people coordinate the clipboard and the wearable camera to maintain the pen and paper inside the camera's field of view while writing? The study was also intended to elicit user feedback and gather design requirements. The findings from the WoZ study informed the design of PaperPal, a wearable smartphone assistant for non-visual interaction with paper forms in more general scenarios than only a stationary desk, such as portable clipboards.

There are several unique aspects to the design of PaperPal. First, its novel 3D-printed attachment transforms a conventional smartphone into a wearable device with a mechanism to adjust the camera angle with one hand. Second, PaperPal is flexible to where it can be used: stationary tables as well as non-stationary surfaces, specifically, portable clipboards. Third, PaperPal enables users to write without having to use their signature guides – a key requirement that emerged from the WoZ study. Fourth, PaperPal leverages real-time video processing techniques to track the paper and pen and accordingly provides appropriated audio feedback. Lastly, both reading and writing are tightly integrated in PaperPal, with users being able to easily switch between them while accessing different items on the document. Our evaluation with 8 blind users showed that PaperPal could successfully assist people who are blind to fill in various paper forms, such as bank checks, restaurant receipts, lease agreement, and informed consent forms. They *independently* filled out these forms with an accuracy reaching 96.7%. We summarize our contributions as follows:

- The results of a wizard of OZ study with blind participants to uncover requirements for independently interacting with paper forms in portable settings.

- The design of a novel 3D attachment that can turn a smartphone into a wearable with adjustable camera angle. This can also be used for other wearable vision-based applications that require adjustment of the camera angle.

- The design and implementation of PaperPal, a new smartphone application, to assist blind users to independently read and

fill out paper forms both on flat tables as well as portable clipboards.

- The results of a user study with blind participants to assess the efficacy of PaperPal in filling out various paper forms.

Following WiYG [28], we also assume annotated paper forms. As mentioned earlier, there exist smartphone applications and known techniques for document image capture by blind people and for automatic annotations. While the annotation problem is orthogonal to the design and implementation of the user interface explored in this paper, in Section 5.11 we describe our experiences with automated annotation of paper forms and discuss its envisioned integration in PaperPal to realize a fully automated paper-form-filling assistant.

## 2 RELATED WORK

The research underlying PaperPal has broad connections to assistive technologies for reading and writing on paper documents, particularly for blind people, 3D printed artifacts and image acquisition and processing in accessibility. What follows is a review of existing research on these broad topics.

*Reading and Writing*: For well over a century, Braille has been the standard assistive tool for reading and writing for blind people. It is a tactile-based system made up of raised dots that encode characters. The use of braille has been declining in the computing era which ushered a major paradigm shift to digital assistive technologies [48]. Examples of digital technolgies for reading printed documents include some CCTVs [63] and Kurzweil Scanner [3] which reads off the text in scanned documents.

The smartphone revolution has witnessed a surge in mobile reading aids. Notable examples include the KNFB reader [2], SeeingAI [8], Voice Dream Scanner [26], Text Detective [14], and TapTapSee [60]. The smartphone-based solutions (e.g., [47]) as well as other hand-held solutions (e.g., SYPOLE [30]) require the user to position the camera for getting the document in its field of view. In recent years, wearable reading aids are emerging (e.g., finger reader [56], Hand Sight [58], and Orcam [7]). Although finger-centric wearables such as [56, 58] do not require positioning of the camera, the drawback is their interference with writing. Reading paper documents using crowd sourced services is another option for blind people (e.g., be my eyes [1] and Aira [11]). These have the obvious drawback of lacking privacy.

In contrast to reading aids, research on assistive writing on physical paper is at a nascent stage. A wizard of OZ study to explore the kinds of audio-haptic signals that would be useful for navigation on a paper form was reported in [17]. In this study, the form was placed on a flat table and the wizard generated the audio-haptic signals that was received on a smartwatch worn by the participant.

A recent paper describes WiYG, a smartphone-based assistant for blind people to fill out paper forms [28]. In WiYG the user places the phone on a stationary table in an upright position using a 3D-printed attachment. The paper form is placed on the desk in front of the smartphone. The user slides the signature guide over the paper form to each form field, guided by audio instructions provided by the smartphone app. To write into the form field the user uses both the hands, one to keep the signature guide in place over the form field and the other hand to write into it with the pen. As mentioned earlier in Section 1, WiYG provides no readouts of the text, simply steps through each form field and can only be used with a flat table where both the paper and the phone are kept stationary. The PaperPal system described in this paper integrates both reading of the document's text and writing in the form fields. It has the capability to operate on both stationary tables and portable clipboards.

*3D Printing in Assistive Technologies*: The increasing availability of 3D printers has increased the potential for rapid 3D printing for assistive technology artifacts [20, 38]. [31] shows that it is feasible

for blind users to do 3D printing of models by themselves and [23] list organizations that use 3D printing tools to serve people with disabilities. Other examples of 3D printing applications are custom 3D printed assistive artifacts [22, 35], 3D printed markers attached to appliances [33], and applications in accessibility of educational content [19, 21, 24, 37], graphical design [46], and learning programming languages [39]. 3D printing is also used to convey visual content [59], art [25], and map information [55] to blind people. Interactive 3D printed objects is yet another way 3D printing is utilized for accessibility [51–53]. Other examples of 3D printing include generating tactile children's books [40, 57] to promote literacy in children. In addition, [41] studies how children with disabilities can use 3D printing. In [36] it is mentioned that children with disabilities can also utilize 3D printing in the context of DIY projects. 3D printing is also used to utilize already existing technologies (e.g., making wearable smartphones [44]). In this paper we utilize 3D printing to design a phone case and a pocketable attachment to turn a smartphone into a wearable that allows the camera's angle to be adjusted.

*Image Acquisition and Processing in Accessibility*: Accessible image acquisition tools such as [15, 43, 62] instruct blind users to position the camera at the correct angle and distance from the target for capturing an image. The work in [32] illustrates the practical deployment of such tools in an assistive technology for image acquisition by blind people. In terms of capturing images of paper documents, assistive reading apps, namely, SeeingAI [8], KNFB reader [2], and voice dream scanner [26] demonstrate that blind people can independently use the apps' interface to direct the smartphone camera on the paper document and capture its image.

The post-processing of the document image is a well-established research topic and can range from local OCR processing [12] to other computer vision methods such as document segmentation [13, 29] to form labeling techniques [9, 10, 49, 61].

Another topic related to camera-based assistive technologies is the use of visual markers for tracking objects in the environment. For example, in [27] different types of visual tracking methods are studied to make shopping easy for blind people. The work in [45] studies color-coded markers for use in a way finding application for blind people. Visual markers are especially beneficial when computer vision methods do not provide satisfactory accuracy. Examples of assistive technologies that utilize visual markers are [28, 54, 55]. In PaperPal we also use visual markers to track the tip of the pen and the paper. To track the latter, PaperPal uses visual markers similar to the ones used for tracking the signature guide in [28]. To track the pen, PaperPal uses visual markers attached to a 3D printed pen topper, inspired by previous work on pen tracking that also use visual markers [21, 64–66].

## 3   A WIZARD OF OZ PILOT STUDY

To the best of our knowledge, there is no previous research on how blind people write on paper documents attached to non-stationary surfaces, namely, portable clipboards. To this end, we did a pilot study to assess the feasibility of an assistive tool that uses a wearable device for filling paper forms attached to clipboards. In the study these specific questions were explored: (1) How do blind people write on non-stationary surfaces like clipboards with a wearable? (2) How do blind people coordinate their hand and body movements to keep the pen and paper within the camera's field of view? (3) What is the most suitable on-body location for a wearable camera between the proposed locations of head vs. chest. In addition, the study was also intended to gather requirements for the wearable camera attachment. Details of the study follows.

### 3.1   Participants

Eight (8) participants (3 males, 5 females) whose ages ranged form 35 to 77 (average age 50) were recruited for the pilot study. All participants were completely blind; all knew how to write on paper;

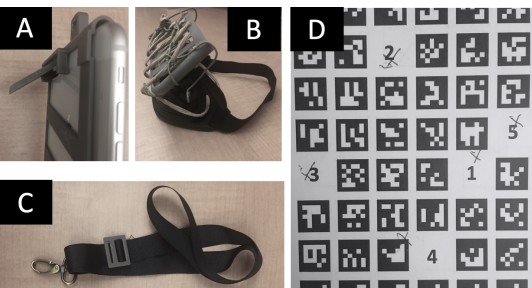

Figure 2: The Apparatus for the pilot study. A: The phone case with reflective mirror in front of the camera to be worn on the chest using the lanyard in C. B: Ski goggles for wearing the phone as glasses. D: Paper with Aruco markers where users mark '×' in the numbered form fields.

none had any motor impairments that would have affected their full participation in the study.

### 3.2   Apparatus

The study used a standard ball point pen, a portable clipboard, and a credit-card sized signature guide. Each form, printed on standard letter-sized paper, had 5 randomly placed equal-sized fields, with the same distance between consecutive fields. The wizard used a Nexus phone to send instructions to an iPhone8+. Participants wore the iPhone8+ on two on-body locations using two holders. The first holder had the iPhone attached to Ski goggles. Participants wore this on the head akin to smart glasses – Wearable$_{head}$ (figure 2 B). Using the second holder, the participants wore a lanyard around their necks with the phone rested on their chests. The holder had a reflective mirror to redirect the camera's field of view and served as the wearable on the chest – Wearable$_{chest}$ (figure 2 A,C).

### 3.3   Study Design

In this within-subjects study every participant filled out a total of 4 random forms corresponding to four different conditions namely, <Wearable$_{head}$, form on desk>, <Wearable$_{head}$, form on clipboard>, <Wearable$_{chest}$, form on desk>, and <Wearable$_{chest}$, form on clipboard>.

A total of 32 forms were filled out (8 participants, 4 conditions). The order of the four form-filling tasks was randomized to minimize the learning effect. The wizard app was used by the experimenter (i.e., the wizard) to manually direct the participant to the form fields by sending directional audio instructions such as "move left", "move right", and so on. The participant's phone had an app to track the paper based on the markers that were printed on the paper (see figure 2 D). The participant's phone was also instrumented to gather study data throughout the duration of each study session which was also video recorded.

The accuracy was measured as the percentage of overlap between the annotated rectangular region of a given form-field (a priori) and the rectangle enclosing the participant's written text in that field (annotated after the study) [28].

### 3.4   Procedure

Each form filling task began with the wizard directing the participant to go to the first form field. We regarded this as the initialization phase and discounted it from our measurements so as to exclude any confounding variables that might arise due to starting off from a random position. Upon reaching the form field the participant would initial the field with a "×" using the signature guide. If at any time during this process the paper disappears from the camera's field of view, the iPhone app would raise a 'paper not visible' audio alert. In response the participant would make adjustments by shifting the paper on or the wearable to bring it back into focus, which gets

acknowledged by the 'paper is visible' shout-out by the app. The participant received the navigational instructions only when the paper was visible by the camera. The experimenter monitored the participant's navigational progress and sent audio directions in real time to guide the user's signature guide to each form field. At the conclusion of the session, users would compare and contrast writing on the desk vs clipboard, the wearable's location on the chest vs the head, and other experiences, in an open-ended discussion.

## 3.5 Key Takeaways

*Chest vs. Head location for the Wearable*: 6 out of 8 participants preferred the chest wearable, one participant preferred the head location and one participant had no preference. With the camera on the head, the paper went out of focus far more often - by a factor of 4 - than when it was worn on the chest.

The differences in percentage of overlap among the 4 conditions were found to be statistically significant (repeated measures ANOVA, $F_{3,124} = 5.32, p = 0.002$). Pairwise comparisons with Bonferroni correction showed that under the "head, clipboard" condition the field overlaps are significantly less than when user wears the phone on the chest and writes either on desk ($p = 0.011$) or clipboard ($p = 0.019$). This suggests that when wearing the phone on the chest, user can better write on the correct location for both papers that are placed both on the desk and clipboard.

These are excerpts of select participant regarding the (1) Wearable$_{head}$: "Looking downward is not comfortable and this task requires a lot of looking down." and (2) Wearable$_{chest}$: "Around the neck is more comfortable and you can focus on the direction of the paper and hand.". Overall, the chest location was better suited for placement of the wearable, and was adopted in the design of PaperPal.

*Writing Surface*: Unsurprisingly, all participants deemed writing on the desk was easier than on the clipboard. However, pairwise comparisons did not show any statistically significant difference in the accuracy or form fill-out time when only the paper placement variable changed from desk to clipboard. During the discussion, all participants mentioned real-life situations where they had to use clipboards.

*Navigational Differences*: On desks, we observed that the user moved the signature guide with one hand while using the other hand to feel the edge of the paper as a means to get a sense of the relative orientation of the signature guide w. r. t. the orientation of the paper. Such a behaviour was also reported in [28]. On the other hand, when holding the clipboard, participants could not feel the paper's edge in the same way as they did on flat desks. In fact, we observed that participants had difficulty moving the signature guide on a trajectory aligned with the paper's orientation. Despite this, the wizard was able to adapt the instructions to lead the participant to the target fields.

*Reflective Mirror*: There was a lot of variability in how participants held the clipboards. This means there is no one perfect angle for attaching the mirror to the holder that can cover all these variations. A wide angle camera increases the likelihood of the paper staying in its field of view. However, this will require the use of a large sized reflective mirror which is not practical. Thus, a smartphone holder without a reflective mirror whose orientation can be adjusted to position the camera, is desirable for a wearable on the chest.

*Signature Guide*: When using the clipboard participants had to hold the clipboard throughout the interaction process. On the other hand, writing with the signature guide requires both the hands. This became a difficult juggling act for the participants. These difficulties are reflected in the feedback of all the participants (e.g.,: P5 mentioned "specially you have to pickup your pen while using signature guide"). Hence using signature guides with clipboards is not an option.

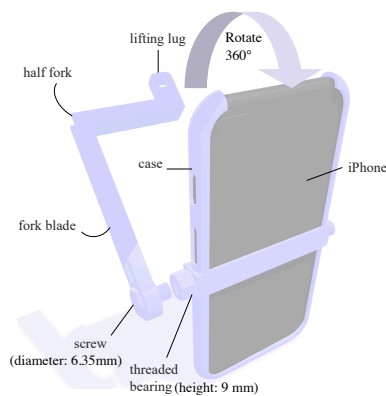

Figure 3: The iPhone holder. The L-shaped half fork can be attached to the phone case by rotating the screw inside the threaded bearing. When tightened the L-shaped half fork can rotate.

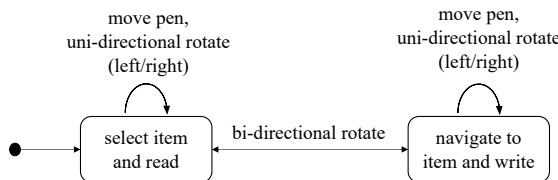

Figure 4: PaperPal's interaction. The interaction automaton.

## 4 THE PAPERPAL WEARABLE ASSISTANT
## 4.1 Design of 3D Printed Phone Holder

Informed by our pilot study, we designed a 3D-printed holder to convert an off-the-shelf smartphone (iPhone 8+) into a hands-free wearable on the chest. In addition, the holder design had to meet these requirements: (a) Support one-handed tilting of the phone to different angles so that differences in how the clipboard is held by different users can be accommodated. They will hold the clipboard with one hand and appropriately adjust the angle of the phone with the other hand to capture the paper in the camera's field of view; (b) Use minimal number of component pieces that are compact enough to fit in a pocket; and (c) Ease of assembly/disassembly.

These requirements led to the design of the *Adjustable iPhone Holder* shown in fig 3 which went through several rounds of experimentation. This design is comprised of two pieces: The first piece is a phone case that has a small threaded bearing on the side. The second piece is a L-shaped half fork which facilitates tilting of the phone. This piece can be attached to the phone case by rotating the screw into the threaded bearing and can be rotated 360 degrees. A lanyard is attached to the lifting lug to wear the holder with its phone around the neck. The user can change the lanyard ribbon's length. This L-shaped fork can be rotated by the user to adjust the tilt angle of the wearable phone. Furthermore, the tilt angle can be adjusted so that it can support upright placement of the phone on a desk. Thus it can operate both as a wearable as well as serve as a stationary holder for writing on a flat desk.

## 4.2 PaperPal: An Operational Overview

The PaperPal system runs as an iPhone app. The user interacts with the items of the paper, namely, text segments, form fields and their labels, by moving the pen over the paper like a pointer and making gestures with the pen. Two types of gestures are used: (1) unidirectional rotate left or right of the pen around its longitudinal axis, and (2) bidirectional rotate made up of two rotations done consecutively in the opposite directions.

PaperPal's response to the user's pen movements and pen gestures is governed by a two-state interaction automaton shown in Figure 4.

The application starts in the "select item and read" state which is inspired by the smartphone screen reader interface. In this state the user can move the pen like a pointer to simultaneously select an item and hear an audio readout of the item that is associated with the location pointed by the pen. This interaction is analogous to the "touch exploration" on the smartphone screen reader.

The unidirectional rotate left (right) selects the previous (next) item on the document and its content is read aloud. This interaction is analogous to "swiping" on the smartphone screen reader.

The bidirectional rotate switches between the two states of the application namely "select item and read" and "navigate to item and write".

The "navigate to item and write" state handles two situations: (1) If the item selected is a text segment it reads aloud the item's text content; (2) if the item selected is a form field it reads aloud its label and generates navigational instructions to direct the user's pen to the location of the field. No readouts of any intermediate items take place when a user is being navigated to a form field. Upon reaching the field it reads out the label of the form field once more to refresh the user's memory and directs the user to write in the field, alerting the user when the pen strays out of the field and giving instructions on how to to move the pen back into the field and continue writing. In this state the user can do a bidirectional rotate at any time to "move to the select item and read" state or continue in the current sate and continue on to the other form fields or other items via unidirectional rotate gestures.

### 4.3 PaperPal implementation

PaperPal uses the phone's camera to observe the user's actions. The application is implemented as an iOS app and uses OpenCV library [6] for real time video processing. Specifically, PaperPal: (a) tracks the physical location of the pen tip over the paper, and (b) detects pen gestures namely uni-directional and bi-directional rotates. The pen location and gestures determine the audio responses, namely, text readouts, navigation and writing instructions, that are generated in real time. Figure 5 is a high-level workflow of the process.

#### 4.3.1 Visual Markers

To enable accurate tracking of the pen and paper, Aruco markers [50] with known size are used for paper and pen tracking.

The paper tracker is a credit-card sized rectangular card (85.60mm × 53.98mm) wrapped by an Aruco board of 24 markers. It has a narrow diagonal groove that serves as a tangible guide for attaching the the paper into the card. The user slides the paper's upper left corner into this groove.

The pen tracker is a cube-shaped pen topper with Aruco markers affixed to each face of the cube. It can be easily attached to any regular ball point pen and is resilient to hand occlusions.

#### 4.3.2 Locating Pen Tip on Paper

For each image frame containing the pen and paper, two transformations $H$ and $P$ are estimated: (1) $H$ is a homography transformation that maps each image pixel to its corresponding location on the paper's coordinate system. This is estimated based on the paper tracker via the DLT algorithm [34], and (2) $P$ is a projective transformation [34] between any 3D location in the pen tracker's coordinate system and their corresponding 2D image pixel. $P$ is comprised of the intrinsic camera calibration, which is measured once for the camera, and the extrinsic camera calibration which is estimated based on the pen markers with the EPnP method [42].

For a pen that is touching the paper, PaperPal starts with the physical location of the pen tip whose distance is a constant w. r. t. the pen tracker coordinate system, and applies $P$ followed by the $H$ transformations to estimate the pen tip location on the paper's coordinate system.

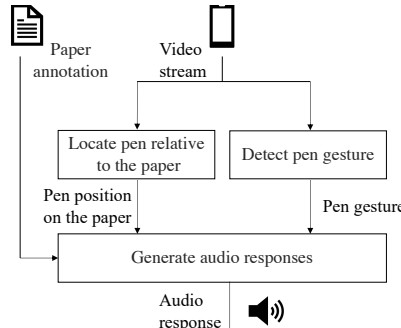

Figure 5: PaperPal workflow.

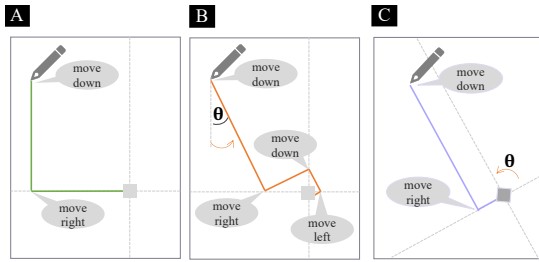

Figure 6: A: rectilinear path along the paper's coordinate system, B: non-rectilinear path along the paper's coordinate system, and C: the rectilinear path on the rotated paper's coordinate system.

We only estimated the pen location if the pen tip is close to the paper. To this end, we developed a heuristic based on the observation that in PaperPal as the pen moves away from the paper it gets closer to the camera. specifically, in the image, the observed size of the pen markers should not be more than twice the size of the paper markers. The criteria was based on experimenting with various thresholds and the one candidate with the lowest average re-projection error [34] was selected. Furthermore we remove outlier pen tip locations whose distance from the previous observed pen tip is more than 18 mm on the paper. This threshold was selected based on the fastest pen movements that could be captured using the pen's visual markers.

#### 4.3.3 Detecting Pen Gestures

A previous work on pen rolling gestures had shown that when writing on a paper, unintended pen rotations were observed with high speeds and small angles [16]. We used this insight to increase the duration of intended rotation gestures and thus distinguish them from unintended ones.

To this end, through experimentation the duration for intended gestures, denoted $T_{gesture}$, was set to 600ms. Rotation gestures are performed with the pen tip on or close to the paper surface. To detect the rotation gestures we choose a 3D point $R$ w. r. t. the pen coordinate system that is close to the pen tip. We find $R$'s corresponding 2D location $r$ on the paper coordinate system using the same process that was used for estimating the position of the pen tip in Section 4.3 above. To detect a rotational movement along the longitudinal axis of the pen, the angle of $r$ relative to the tip of the pen is measured for each frame using simple trigonometry. We record the direction of the rotational angle (left/right) between each two consecutive frames. If in the time window of $T_{gesture}$, the majority (over 90%) of pen rotations are in the same direction (left/right), the rotation gesture in the corresponding direction is detected. For detecting bidirectional rotations, first note that it involve two quick rotation in both direction. A bidirectional rotate is detected if within the sliding time window of $T_{gesture}$, majority (over 90%) of rotations

in one half in one direction and in the opposite direction in the other half.

### 4.3.4 Generating Audio Responses

PaperPal responds to the user's pen movements by generating four kinds of audio responses: coordination alerts, audio readouts, writing and navigation instructions.

Coordination alerts: Alerts when the paper and/or the pen move out of the camera's field of view. To avoid needless alerts for momentary movements out of the field of view, we alert when the pen and paper are not in field of view for a duration that is longer than 2 seconds.

Readouts: The textual content of an item selected by the user is readout to the user in audio.

Writing instructions: While writing is in progress, instructions to maintain the pen within the field is given whenever the estimated pen tip falls out of the rectangular boundary of the field. For example, if the user's pen position has strayed above (below) the field the user is guided back to field with a "Move down (up)" instruction.

Navigation instruction: Navigational instructions consists of four basic directives namely up, down, left and right. With these four directives the user is guided to any field on the paper. In [28] a simple navigation algorithm was used to guide the user along a rectilinear path that corresponded to the Manhattan distance between the pen and field, see figure 6 A. Recall that our pilot study revealed that the user's navigational movements, in response to the audio instructions can deviate from the intended axes when the paper is placed on a clipboard. Figure 6 B, demonstrates how the user's pen tip trajectory can deviate from the expected path along the paper's coordinate system with simple rectilinear navigational instructions.

To address this problem we estimate the deviation angle of the pen tip's trajectory w.r.t. the paper's coordinate system. To compensate for this deviation, we rotate the paper's coordinate system to the same degree but in the opposite direction, so that the pen tip's trajectory is aligned with the transformed axes – see figure 6 C. The navigation directives are generated w.r.t. the transformed axes. Observe that the pen tip trajectory now follows a rectilinear path w.r.t. the transformed axes. To estimate the deviation angle, we use the pen tip's estimated location $t$ on the paper and find the angle between $t$ and the intended axis (horizontal axis when the navigation instruction is left or right, and vertical when the navigation is up or down). To avoid noise and jitters of the transformed axes, the deviation angle is averaged over a sliding window of one second.

## 5 EVALUATION

We conducted an IRB-approved user study of PaperPal to evaluate its effectiveness as a form-filling assistant for blind people. To this end, the study was designed to answer the following questions: (a) How accurately can users fill out forms in terms of writing on the correct location? (b) How long does it take to fill out forms? (c) What is the overall user experience of using PaperPal to fill out paper forms of different sizes, layouts, and texts?

### 5.1 Participants

Ten fully blind participants were recruited. However, two participants could not attend and the study was conducted with the remaining eight participants whose ages ranged from 32 to 63 (average = 47.88, std = 12.16, 4 females and 4 males). Note that 4 out of the 8 participants were also part of the WoZ pilot study discussed in Section 3. Table 1 is the demographic data of the participants. The participants were compensated $50 per hour. All the participants were right handed, and none had any motor impairments that impeded their full participation in the study. All the participants (except P5) were familiar with braille and all of them affirmed that they knew how to write on paper. All participants stated that in real-life they always asked a sighted peer to do the form fill out for them except for affixing their signatures. For that, they were led by the sighted peer to the signature field where they would sign by themselves.

### 5.2 Apparatus

The PaperPal application was running on an iPhone 8+. The 3D-printed holder, lanyard, paper tracker, pen tracker, a regular ball point pen and a clipboard were provided to the user, see figure 1. Finally, each participant was given 4 paper forms to fill out.

Forms: The forms were selected to have different properties and to reflect realistic scenarios. Specifically, the forms were:

- (F1): A regular-size check that consists of six fields namely *pay to the order of*, *date*, *$*, *dollars*, *memo*, and *signature*.

- (F2): A restaurant receipt that consists of three fields namely *tip*, *total*, and *signature*.

- (F3): A template for a lease agreement that consists of the following six fields: *landlord's first name*, *landlord's last name*, *tenant's first name*, *tenant's last name*, *landlord's signature*, *date*, *tenant's signature*, and *date*.

- (F4): An informed consent form that requires the participant to fill out four fields which are *full name*, *date of birth*, *participant's signature*, and *date*.

The two forms (F1 and F2) are quiet similar to the ones used in the evaluation of the WiYG [28]. Two additional forms were selected to evaluate more complex forms in terms of the number of fields (F3) and text items (F4) – see Figure 7. These forms have different paper sizes, orientation, and form layouts. Specifically, F1 and F2 forms are smaller than the standard letter size pages used in F3 and F4. The fields in F2 are vertically aligned and placed below one another, F3 fields are placed on horizontally on a table-like layout, and F1 and F4 have more complex layouts.

### 5.3 Design

The study was designed as a repeated measures within-subject study. Each participant was required to fill out each of the 4 forms (4 tasks) with PaperPal in a counterbalanced order using a latin square [18]. The task completion time was the elapsed duration from the moment the pen was detected over the paper for the first time and the moment the user finished writing on the last form field. Accuracy was measured as the percentage of overlap between the ground truth annotated rectangular region of a given form field (a priori) and the rectangle enclosing the participant's written text for that same field – see figure 7.

### 5.4 Procedure

To start with, we draw attention to the circumstances surrounding the study. It was conducted after the gradual re-opening of businesses shut down due to the COVID-19 pandemic. Consequently, the study procedure was adapted to follow CDC recommended safety measures. Specifically, both the participant and the experimenter wore face masks and kept the recommended social distance from each other. Therefore, the experimenter relied on verbal communications instead of physical demonstrations to conduct this study.

Each session began with a semi-structured interview to gather demographic data, reading/writing habits, and prior experiences with assistive smartphone apps ($\approx$20 minutes) – see table 1.

Following this step, the participant was instructed on how to set up the PaperPal apparatus. Towards that, the participant was asked to pick up each piece of the apparatus and the experimenter would provide a verbal description of the pieces, after which the participant began assembling the pieces with the experimenter giving step-by-step assembly instructions until the participant was able to attach the paper tracker to the paper, the pen tracker to the pen, the L-shaped fork to the phone case, clip paper to the clipboard and the lanyard to wear the phone around the neck. After that, the experimenter described the user interface. The participant was asked to practice reading and writing with PaperPal with a set of test forms that were

Table 1: Participants demographic and habits regarding braille, writing, and smartphone applications (SG stands for signature guide).

| ID | Pilot study | Age (Sex) | Diagnosis (Light perception) | Braille usage (Level) | Braille scenarios | Writing usage (Level) | Writing scenarios | SG Own (Carry) | Smartphone (experience) | Smartphone apps for papers |
|---|---|---|---|---|---|---|---|---|---|---|
| P1 | yes | 34 (F) | retrograde optic atrophy (yes) | daily (advanced) | papers at work, at the library | daily (beginner) | doctor's office, legal forms, banks | yes (always) | iphone 10 S (advanced) | seeingAI, KNFB reader, voice dream reader, voice dream writer |
| P2 | yes | 63 (F) | Acute congenital glaucoma (no) | daily (advanced) | taking notes for myself | rarely (beginner) | Leaving notes for sighted peers | yes (always) | iPhone 8 (advanced) | be my eyes |
| P3 | no | 32 (F) | medical malpractice (no) | daily (beginner) | elevators, remote control | weekly (advanced) | doctor's office, checks | no | iPhone 10 (advanced) | None |
| P4 | no | 55 (M) | retinal detachment (no) | monthly (advanced) | elevators, mails | weekly (beginner) | timesheet signature, checks, legal documents | no | iPhone (advanced) | KNFB reader |
| P5 | yes | 54 (M) | glaucoma (no) | - | - | weekly (beginner) | shopping receipts, credit card bills | no | iPhone (beginner) | seeingAI, tap tap see |
| P6 | yes | 46 (F) | retinitis pigmentosa (yes) | never (beginner) | elevators, mails | monthly (beginner) | legall documents, doctor's office | no | iPhone 8 (advanced) | seeing AI |
| P7 | no | 38 (M) | optic atrophy, retinitis pigmentosa (yes) | monthly (advanced) | reading documents | daily (advanced) | documents at work, taking notes for sighted peers | yes (often) | iPhone 11 pro (advanced) | KNFB reader, seeingAI, Aira, voice dream scanner |
| P8 | no | 61 (M) | retinitis pigmentosa (yes) | daily (advanced) | elevator, calendar | daily (advanced) | timesheet signature, checks, legal documents | yes (always) | flip phone (beginner) | None |

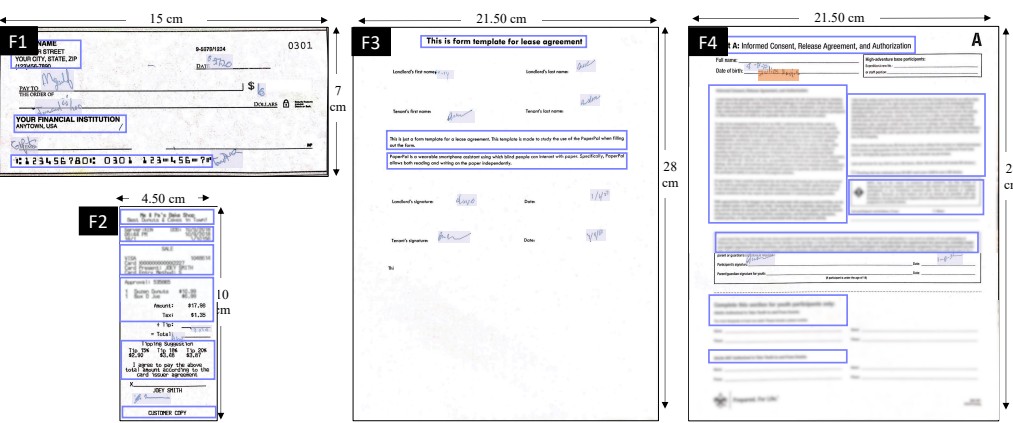

Figure 7: The four forms used in the user study. Note that the scale of the images does not represent their relative size (refer to the dimensions in the figure). The participants' hand writings are annotated with blue (brown) as correct (incorrect) by human evaluators.

different from those used in the study. During the practice the experimenter observed the progress of the participant and intervened with instructions and explanations as needed. The entire process of assembling and practicing use of the application took about an hour.

The participant was next asked to fill out the four forms F1, F2, F3 and F4, followed by a single ease question. A maximum of 10 minutes per form was allocated. An open-ended discussion with the experimenter took place upon completion of the tasks. The entire study session per participant lasted 2.5 hours, with the experimenter making notes throughout the video recorded session.

### 5.5 Results: Task Completion Time

The task completion time is indicative of the efficiency of PaperPal as a form-filling assistant for blind people. On average, the total time spent to fill out forms F1 to F4 was 169.38, 73.91, 229.824, and 164.84 seconds respectively. The task completion time is divided into: (a) Navigation time, which is the time taken to navigate the to the target field; (c) Writing time, which is the time taken by the participant to fill in the field; (d) Coordination time, which is the time taken by the participant to bring the paper back in the camera's field of view. Figure 8 shows both the navigation time and writing time for each field.

### 5.6 Results: Assembly Time

The holder assembly time was includes time spent to attach the L-shaped half fork to the phone case and wear the phone around the neck. All participants were able to assemble the holder with an average time of 16.55 seconds (std 4.33 seconds). In addition, the average time to attach the paper tracker card to the top-left corner of an A4 page was 18.75 seconds (std = 6.45 seconds). One difficulty that arose was that most participants were wearing protective gloves which made it difficult for them to attach the paper to the paper tracker card.

### 5.7 Results: Form Filling Accuracy

*Overlap Percentage*: Recall that the overlap percentage is defined as the percentage of the rectangular bounding box enclosing the participant's writing that falls inside the ground truth rectangular region of the field, see figure 7. The average percentage of field overlap for forms F1 to F4 was 61.84%, 66.02%, 87.69%, and 64.23% respectively.

Out of all the form fields in this study (8 participants $\times$ 21 fields = 168) participants attempted to fill out 156 of them. Out of these 156 fields, 117 (75%) of them had an overlap region of 50% or higher. Figure 8 (Field Overlap) shows the overlap percentages.

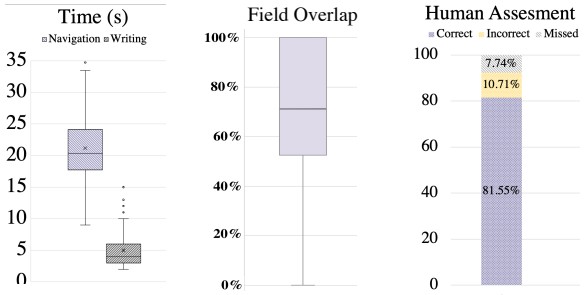

Figure 8: left: navigation time and writing time per field. middle: accuracy in terms of the field overlap percentage. right: human assessment of the filled-out fields

*Human assessment*: We asked three human evaluators to assess whether each of the form fields were correctly filled out by the participants. The final verdict was rendered via majority voting. The inter-annotator agreement was high (Fleiss' = 0.77). Figure 8 (Human Assessment) shows the percentage of fields that were deeded as correct (81.55%), incorrect (10.71%), or missed (7.74%).

The human assessment of form-filling accuracy for forms F1 to F4 was 88.09%, 69.56%, 96.77%, and 85.71% respectively. Note that human assessment shows higher accuracy compared to the average overlap. Akin to [28], this phenomenon is due to the fact that even when the written content is not perfectly inside the form field, human annotators still counted it as acceptable.

## 5.8 Results: PaperPal vs. WiYG

The standard check (F1) and receipt (F2) forms were similar to the forms used in WiYG's user study in that it also used a check and receipt of similar size and layout and identical set of fields. Although the differences in the setup and study participants preclude a rigorous comparison of the performance between PaperPal and WiYG, we can still get a sense of the differences in their performance via an informal comparison shown in table 2. This comparison suggests that in spite of the complexities of writing on a non-stationary clipboard with a wearable, users could fill out forms in a shorter time without compromising accuracy with PaperPal.

## 5.9 Results: Subjective Feedback

We administered a single ease question to each participant to rate the difficulty of assembling the holder and completing each form, on a scale of 1 to 7 with 1 being very difficult and 7 being very easy. The median rating for holder assembly was 7 which suggested that the assembly process was viewed as being easy. The median rating for forms were F1: 6, F2: 5, F3: 4, F4: 3.5. In the open-ended discussion, participants mentioned that filling out forms that had long text content (such as F4) or had more fields were more difficult.

All participants liked the fact that PaperPal lets them work with paper documents independently – quoting P5:" I like the ability to fill out my own forms and checks". All of them mentioned that PaperPal fills an unmet need for preserving privacy when filling out forms and affixing signatures. They all appreciated the integrated reading and writing feature in PaperPal as they got to hear what was in the form that they were filling out prior to affixing their signatures.

## 5.10 Discussion

In the user study participants missed filling 12 out of 168 fields. We observed most of the missing cases were associated with F3's two *date* fields, which were next to each other and had the same label, causing ambiguity. The last two fields in F4 were also missed by some participants because the long text gave them the false impression that there were no more fields left in the form. One way to address this in future work is to apriori notify the user of the total number of form fields.

All participants mentioned that doing several consecutive rotate gestures required re-adjustments to their grip on the pen. Gestures with subtle finger movements such as finger flicks and taps on the pen are possible alternatives that can address this problem. This will require the use of computer vision recognition algorithms to detect subtle finger movements and is a topic for future research.

A unique aspect of PaperPal is that users can simply work with the paper and pen without having to hold any other objects like the phone in reading apps [2, 8] or signature guide while filling out paper forms as in WiYG. Finally, assembly of the 3D-printed attachment, attaching the trackers, and wearing the phone were all done independently by the study participants, affirming that the design of the apparatus associated with PaperPal was highly accessible for blind users.

The results of the study showed that blind participants were able to fill out forms in a few minutes (ranging from 1 min and 23 sec to 3 mins and 83 sec) with high accuracy, measured in terms of the average overlap percentage, which was more than 60% for all the forms. In addition, accuracy of the filled out form fields as judged by humans was also as high as 96.77%.

## 5.11 Future Work

*Use of Markers*: The PaperPal's accuracy depends on accurate tracking of pen and paper. Which is why it is done with visual markers attached to these objects. Eliminating these markers is a challenging open computer vision research.

*Document Annotation*: While the focus of this paper has been the accessible HCI interface for filling paper forms with a wearable, we envision a front-end to PaperPal consisting of an app like KNFB reader or voice dream scanner to acquire the image of the form document which subsequently will be dispatched to the augmented AWS Textract services for automatic annotation of the form elements. We conducted preliminary experiments with this service. To this end, we took pictures of the 4 forms used in the study (Section 5) with the voice dream scanner app. These images were rectified using the "image to paper" transformation (section 4.3) and these rectified images were processed by AWS Textract augmented with human-in-the-loop workflow. Out of the 21 fields in the 4 forms, 17 fields and their labels were detected correctly by Textract and 2 were erroneously recognized and were marked as such through intervention via the the human-in-the-loop workflow. Integration of this process to PaperPal and its end-to-end evaluation is a topic of future work.

## 6 Conclusion

PaperPal is a wearable reading and form-filling assistant for blind people. Wearability is achieved by transforming a smartphone, specifically an iPhone8+, into a wearable around the chest with a 3D printed phone holder that can adjust the phone's viewing angle. PaperPal operates on both stationary flat tables as well as non-stationary portable clipboards. A preliminary study with blind participants demonstrated the feasibility and promise of PaperPal: blind users could fill out form fields at the correct locations with an accuracy of 96.7%. PaperPal has potential to enhance their independence at home, at work and on the go and at school.

## 7 Acknowledgements

We thank the anonymous reviewers for their insightful comments. This work was supported by NSF awards 1815514, 1805076, 1936027 and NIH awards R01EY030085, R01HD097188.

Table 2: Comparison of PaperPal to WiYG (accuracy: overlap%)

| | Standard Check (F1) | | Receipt (F2) | |
|---|---|---|---|---|
| | average time (s) | average accuracy (%) | average time (s) | average accuracy (%) |
| WiYG [28] | 249.75 (s) | **64.85%** | 91.25 (s) | 63.90% |
| PaperPal | **159.38 (s)** | 61.84% | **73.91(s)** | **66.02%** |

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
