# OpenReview forum: "Towards Enabling Blind People to Fill Out Paper Forms with a Wearable Smartphone Assistant"
_graphicsinterface.org/Graphics_Interface/2021/Conference/Second_Cycle — GI 2021_

### Official Review · Reviewer_TRsM · 2021-04-25
**Novel, but thin contribution. Some correctable issues with methodology**

**Rating:** 6
**Confidence:** 4

**Review:**

This paper presents an exploratory study, the design and prototype for PaperPal, and the subsequent evaluation of the PaperPal prototype, which is aimed at making paper-based forms easier for blind users to fill.

Overall, I found the paper easy to read, and the contribution quite clear. The paper does well in situating itself with respect to prior work, especially with the clear contrast to WiYG, which proposes a very similar approach. I think the paper is novel and adds to the literature. The formative study provides some insight into issues and opportunities with the proposed approach, which serves as nice grounding for the system that follows. The system description and some details are a bit sparse, and the evaluation found that PaperPal was a reasonable improvement over WiYG, though not wildly more impressive. Overall, I think there is novelty here, but the contribution is somewhat minimal across the board (the formative study doesn't provide deep insights, the design and implementation of PaperPal is not significantly novel or challenging academically, and the results do not show substantial improvements)

I found some errors in the analysis, which should be addressed prior to publication:

Running formal NHST (RM-ANOVA) seems excessive with only 8 participants. A visual presentation of this data would be useful (e.g., chart with error bars for each of the conditions). Tukeys tests are not appropriate for non-independent samples. Bonferroni-corrected pairwise t-tests should be run instead (or another suitable measure). The same problem exists with the evaluative study. I don't think the results will change significantly (nor do I think such an analysis is necessary for this type of contribution), so I would not consider this a significant barrier to publication.

Beyond those errors, I would expect a little more detail in the study design as well as the system design.

Why was an 'x' used in place of the participants natural signature or mark?
What was the content of the four random forms in the formative study (content, structure, number and type of questions?).
2.5 hours seems like a a very long time for a study when the total time to complete all forms was approximately 15 minutes, with less than a minute for setup. Add in 20 minutes for the introduction, and where did the remaining ~2 hours go?

Beyond those, the paper could be improved by including charts/graphs that demonstrate their results. Additionally, the figures could be placed closer to their reference in the text which would improve the reading experience.

I think this paper is interesting, I don't think there are any fatal flaws and I hope it gets published in an improved form. I would consider it borderline for Graphics Interface, after the analysis has been corrected and the content clarified.

---

### Official Review · Reviewer_Lsux · 2021-05-03
**An enjoyable design case-study**

**Rating:** 7
**Confidence:** 4

**Review:**

This paper describes PaperPal, a smartphone-based assistant? technique? process? for allowing blind users to independently complete paper forms either on a desk or on a clipboard. The paper includes a wizard-of-oz study to assess where the smartphone should be placed to support desk and clipboard form completion, a description of a holder to allow blind users to mount the smartphone, and a summative study that examines how effective PaperPal is at supporting form filling tasks both on desks and on clipboards.

When I break apart the components of this paper, there isn't much novelty in any individual aspect of the paper. Smartphones are useful for tracking text and documents (we see this with Google translate, for example, where you can hold up your phone and its camera will identify, analyze, and translate text for you in a AR-centric way). We know this already. Fully blind users are unlikely to use head direction to track forms as they don't need to focus their eyes, but body direction makes sense. We can 3D print brackets to hold phones, and design these brackets with two or three assembly points.

However, when I assess this paper, while I don't see any particular novelty in any one aspect of the paper, at an overall level I thought that the execution of the design exercise is useful. My litmus test for a GI paper is whether I could give a paper to my graduate student and have him or her get something valuable out of the paper. From that standard, I think this paper is above the bar. It describes, in a useful way, a case study in how to generalize a pre-existing system that supports form fill on desktop surfaces to a more mobile context where users may have forms on clipboards. It truncates some of the non-HCI aspects of the research (e.g. the computer vision problem) by using fiducial markers so that it can focus on the ability of a wearable smartphone to support blind users in a form filling task. It collects feedback from a system that realizes that process to assess whether or not the design decisions made (the bracket, the interactions) make sense for the target population.

I could go into the details of the paper in greater depth, but, for me, this isn't really the point of this paper. It isn't really about formal experimentation; it's about usability engineering. Having identified a need to generalize a system for additional contexts, the paper takes a relatively straightforward approach to this generalization. The contributions are clear: it is useful to be able to fill out forms independently when blind, and wearing a smartphone on one's chest on a bracket supported by a lanyard works to allow blind participants to do this.

---

### Official Review · Reviewer_FoSb · 2021-05-03
**Documents the user-centered design of a system to enable blind individuals to independently fill out paper forms using a smartphone assistant.**

**Rating:** 8
**Confidence:** 4

**Review:**

This paper documents a well-designed user-centered design process to develop a wearable smartphone assistant to aid blind individuals in individually filling out paper forms. The paper does a good job of motivating the problem and outlining the current state of the art. The design of the system is based on a wizard of oz study with 8 blind participants to guide the design of a wearable device. This system is then implemented both in terms of the software application itself and a 3-D printed attachment to turn a standard smartphone into a wearable device supporting the use-case. The efficacy of the system is then evaluated in a study with 8 blind participants (partially overlapping with the participants in the first study) establishing that individuals were able to mostly fill-out the forms correctly and independently on a range of form types, providing preliminary evidence supporting the system’s success. The paper final identifies remaining challenges (e.g., particular field types that were less successfully completed) and outlines possible solutions.

An understated point that I would like to draw out is that the paper deserves praise for the practicality of the solution. Although the solution may seem a bit clunky (relative to say a novel wearable device), it is practical. It doesn’t require buying fancy hardware (for a relatively infrequent task) and fits within what we might expect most of the target population to minimally possess (an older generation smartphone). The only additional purchase would be the attachment which seems like it could be offered at a reasonable cost and with reasonable portability.

The final evaluation is perhaps the weakest part of the paper, and only because it tries to do too much. Overall, I think it is reasonable in scope given that the paper already makes a substantial systems contribution, but it could have been presented more strongly by dropping the statistical comparison of form types. I’m not sure what the point of this is – do we care that certain forms are faster than others? They weren’t designed to be equivalent, and the results seem to track quite well with length/complexity. Longer forms with more fields took longer to complete… as we would expect.

I would argue that the subjective results of how participants reacted to the system along with descriptive statistics on the success rate best demonstrate the extent to which the system succeeds in achieving its goals of supporting independent form filling. Given that there aren’t existing systems to support this (as established in prior work and the WOZ study, a system comparison isn’t possible. And while a longitudinal field study establishing real world use would be lovely, such an evaluation is above and beyond what should be reasonably be expected to complement a systems contribution. Cutting the inferential analysis could make more space for greater detail on the subjective findings and/or discussion of their interpretation, but I think the paper is acceptable as is.

Overall, I think this paper makes a nice contribution and I believe that it should be accepted to GI 2021.

---

### Meta-Review · Area_Chair_fcCa · 2021-05-07

**Recommendation:** Accept
**Confidence:** 5

**Metareview:**

Overall, reviewers are leaning to positive on this paper, and reviews are largely in agreement that the overall system as case study is worthy of publication. Overall, reviewers agree that:
- the combination of factors in design provides a nice system
- the formative work to guide design results in a system that seem practically useful
- the paper is well written.

There are some minor comments on statistics from one reviewer, but, as this reviewer notes, these aren't a barrier to publication, merely an opportunity for some improvement in presentation of the study.

---

### Decision · Program_Chairs · 2021-05-08

Accept